# Yearly Energy Performance Assessment of Employing Expanded Polystyrene with Variable Temperature and Moisture–Thermal Conductivity Relationship

**DOI:** 10.3390/ma12183000

**Published:** 2019-09-16

**Authors:** Maatouk Khoukhi, Shaimaa Abdelbaqi, Ahmed Hassan

**Affiliations:** College of Engineering, United Arab Emirates University, 15551 Al Ain, UAE; 200734406@uaeu.ac.ae (S.A.); Ahmed.Hassan@uaeu.ac.ae (A.H.)

**Keywords:** energy performance, space cooling, AC capacity, building insulation

## Abstract

This paper investigated the impact of the changes of thermal conductivity of an expanded polystyrene insulation layer embedded in a typical residential building on the cooling effect at different temperatures and moisture contents. The simulation was performed using expanded polystyrene (EPS) in the extremely hot conditions of Al-Ain (United Arab Emirates, UAE) at different levels of density, denoted as low density LD (12 kg/m^3^), high density HD (20 kg/m^3^), ultra-high density UHD (30 kg/m^3^), and super-high density SHD (35 kg/m^3^), and three moisture content levels (10%, 20%, and 30%), compared to dry LD insulation material. The thermal performance of the building incorporating polystyrene with variable thermal conductivity (λ-value) was compared to one with a constant thermal conductivity by assessing the additional cooling demand and capacity due to the λ-relationship with time, using e-quest as a building energy analysis tool. The results showed that, when the λ-value was modeled as a function of operating temperature, its effect on the temperature profile during daytime was significant compared with the use of a constant λ-value. The monthly energy consumption for cooling required by the building was found to be higher in the case of variable thermal conductivity for the LD sample. The yearly average change in space cooling demand and cooling capacity when employing polystyrenes with constant and variable thermal conductivity increased with the increase of the moisture content. Indeed, the highest changes in cooling demand and capacity were 6.5% and 8.8% with 30% moisture content polystyrene.

## 1. Introduction

The building sector is responsible for more than 36% of global final energy consumption, and nearly 40% of total direct and indirect CO_2_ emissions. The energy consumption from building continues to rise due to rapid growth in the building sector. In harsh climates, where industrial activities are not extensive, the building sector contributes around 70% of the total energy requirements, mainly due to the use of AC systems [1].

The building envelope represents an effective boundary and a physical barrier between the internal and external environments [2]. Insulation material is a layer composed of a single material or combination of materials that essentially contributes to the overall thermal performance of the opaque walls [3,4], possessing the characteristic of high thermal resistance, which has the capability to decline the heat flow rate [5], and responding to the external conditions with its specific thermophysical properties [4].

The thermal conductivity of insulation (λ) is generally considered to be a constant in pertinent calculations. This is, however, not true, as the λ-value of a building envelope—a wall, for instance—exhibits variation with the operating temperature and moisture content.

### 1.1. The Effects of Temperature on the Thermal Performance of Insulation Material

Aldrich and Bond investigated the effects of temperature on the thermal performance of rigid cellular foam [6]. Their results showed a significant change in the λ-value with temperature changes. Several studies have reported this dependence in recent years, with the λ-value generally found to increase with temperature and moisture content [7,8,9]. Khoukhi and Tahat also investigated variations in the λ-values as a function of the density and operating temperature of EPS insulation material, as well as the effects of those changes on the cooling load required by buildings [10,11,12].

Recently, Berardi and Naldi [13] investigated the impact of the temperature-dependent thermal conductivity of insulating materials on effective building envelope performance. They concluded that the variation in conductivity in terms of temperature is almost linear for inorganic fiber insulations and some petrochemical insulating materials. However, the variation in conductivity as a function of temperature has been found to be non-linear for blown foam insulation.

### 1.2. The Effects of Humidity Content on the Thermal Performance of Insulation Material

A number of researchers have reported the effect of moisture transfer on the thermal performance of insulation materials [14]. It has been reported that the presence of moisture in an insulation material changes its thermal performance [15]. A reciprocal function between the density and water was presented by Gnip et al. [16]. The presence of liquid in insulation also has a huge impact on the thermal conductivity of the insulation material [17]. Previous results indicate that the accumulation of moisture in building materials leads to an increase in their thermal conductivity or K-value, as well as a decrease in their insulation capacity [18,19,20,21,22].

The thermal conductivity functions of four materials, namely rock wool, fiberglass, extruded polystyrene, and polyisocyanute have been previously created, and dynamic simulations were run for typical construction components of a building envelope [23]. This investigation was done on exterior walls and flat roofs under different climactic conditions in Italy. According to the results obtained, polyisocyanurate demonstrated a larger performance variability with respect to the other materials, highlighting the potential inaccuracies that may introduced in building performance estimation by assumptions about the thermal conductivity of insulation materials.

Recently, several advanced insulation materials have been developed, referred to as dynamic insulation materials (DIMs) [24], which are expected to be useful for many applications in future technology [20]. Recent findings show that the use of DIMs could save up to 17% on the annual cooling and heating energy costs incurred by U.S. office buildings [25]. Similarly, novel adaptive insulation technologies could provide an opportunity to reduce building energy use by modulating heat gains and losses between outdoor and indoor environments [26,27].

The main objective of the present study was to investigate the impact of the change of the thermal conductivity of EPS materials on the heat transfer through a wall assembly for different densities of EPS material, in terms of operating temperature and moisture content. The required space cooling and the yearly average change in space cooling demand and cooling capacity were calculated for both constant and variable thermal conductivity. The difference in space cooling demand and capacity for the whole year at different moisture contents for LD insulation material was assessed accordingly.

## 2. Materials and Methods

### 2.1. Measurment of Moisture Contents

The ability to absorb moisture on the K-values of EPS insulation with different densities has been previously investigated by the authors [1] using a customized apparatus to mimic air moisture transfer. During the experimental measurement, it was noticed that the effect of moisture on HD, UHD, and SHD samples was insignificant, due their impermeability to moisture transfer because of their high density. Therefore, only the LD sample was considered for the current investigation. The best-fit linear relationships between the K-values and moisture content were obtained as below:(1)$$y\;=\;6\;\times 10^{-5}x\;+\;0.0357$$

### 2.2. Heat-Transfer Analysis

The heat-transfer analysis across the wall section was modeled and solved using the ANSYS platform (Version 18, Computer software company, Cecil Township, PA, USA, 2018), adopting Al-Ain climatic conditions, characterized by hot weather in July. The problem was solved transiently, applying the user-defined daily transient equations of the weather conditions. The transient temperature distribution at each node was determined iteratively by the solver, and the average surface temperature was processed. The solution was updated at 1 min intervals: that is, after completing 20 iterations, for a total run time of 24 h.

The total solar energy received by the outer building surface (concrete stucco surface) during the day was calculated using Equation (2) [28]:(2)$$Q_{in}=\sum_{n=1}^{n}G_{i}\times A\times \varphi \times t_{i}$$where *G* is the global solar radiation intensity incident on the surface (W/m^2^), *A* is the surface area of the concrete stucco surface facing south, *φ* is the absorbance coefficient of the concrete stucco, and *t* is the time in hours. Applying (20 × 3) m^2^ as the concrete stucco surface area of the residential building and inserting *φ* = 0.65 into Equation (2), a daily *Q_in_* of approximately 111 kWh was obtained.

The heat losses, when the building envelope was modeled using variable-λ polystyrene with different densities and different moisture contents, were calculated by comparing the resulting indoor temperatures in each case, using the expression below:(3)$$Q_{inner}=\;h_{c}\;\times \;A\;\times \;\left(T_{s}-T_{i}\right)$$where *h_c_* is the convective heat-transfer coefficient (between the inner surface and the interior), *T_s_* is the inner surface temperature, and *T_i_* is the indoor air temperature. The *h_c_* value at the inner surface facing indoors was calculated at 6.5 W/m^2^ °C using Equation (4), assuming free cooling and applying the wind speed ($v_{w}$) for UAE (United Arab Emirates) [29].
(4)$$h_{c}\;=\;3.3\;v_{w}\;+\;6.5\;$$

### 2.3. Building Energy Performance

The cooling energy demand of a typical one-story building (20 m × 20 m × 3 m) located in Al-Ain, UAE (all building characteristics presented in Table 1), with a commonly used wall construction assembly comprising a 200 mm thick concrete block layer, a 50 mm insulation layer, a 13 mm thick interior gypsum board, and a 19 mm concrete stucco at the exterior surface, was numerically simulated using the e-quest program as a building energy analysis tool.

The numerical model enabled the analysis of specified multizone buildings including heating, ventilation, and air conditioning (HVAC) systems, internal loads from people from 10 residents (as an average in local UAE houses), equipment including all the house appliances, and lighting, applying the Al-Ain, UAE hot weather condition. The activities were set to medium level in the morning, low during the day time, and high during the night time, while the set temperature was selected to be 25 °C.

## 3. Results

### 3.1. Measured Weather Data in Al-Ain, UAE

The average ambient temperature and the average hourly and total solar radiation of Al-Ain for a typical day of each month are shown in Figure 1 and Figure 2, respectively. Al Ain is characterized by long and very hot summers and mild winters.

### 3.2. Monthly Energy Demand for Space Cooling

Based on the obtained weather data, the monthly energy demand required for cooling a residential house located in Al-Ain was obtained using constant (c) and variable (v) thermal conductivity of the dry low density polystyrene, and the result is shown in Figure 3.

### 3.3. Yearly Energy Performance

The total yearly average and peak additional energy demand for the air conditioning system to remove the heat from the space by percentage (additional space cool) were obtained for different levels of polystyrene density and different levels of moisture content, as shown in Figure 4 and Figure 5, respectively.

### 3.4. Yearly Cooling Capacity and Required Air Flow

The total yearly cooling capacity and required supplied air flow applying different level of polystyrene densities (Figure 6) and different levels of moisture contents (Figure 7) were calculated.

## 4. Discussion

### 4.1. Monthly Energy Demand for Space Cooling

As the ambient temperature in Al-Ain, UAE increases in the summer season, the energy demand for cooling purposes of residential buildings reaches its maximum during the hot months. The average ambient temperature during the hot months reaches up to 43 °C, while the solar radiation during the same months reaches 650 kW/m^2^ as a total daily.

As a result of such high ambient temperatures, non-uniform monthly energy consumption for cooling purposes was obtained by simulations of the residential building across the year. Indeed, the energy demand was at maximum during the hot months of July and August. Thus, the changes in cooling demand between using constant and variant thermal conductivity reached their maximums in these months. The peak cooling demands for the residential building house in Al-Ain, using polystyrenes with constant and variable thermal conductivity as part of a wall section during July were 4.15 kWh and 4.32 kWh, respectively.

### 4.2. Yearly Energy Performance

Depending on the obtained date and applying the measured weather data in the simulation program, the yearly average and peak additional space cooling percentage employing dry variable thermal conductivity in the wall sections of the residential building were calculated, as compared with constant value of polystyrene thermal conductivity at different density levels (Figure 4) and at different moisture levels (Figure 5) in the hot climate of UAE.

From Figure 4, a low-density polystyrene applied in the wall section of the residential building in hot weather in Al-Ain, UAE resulted in the lowest yearly average of additional space cooling required (0.4%), while using a super-high-density polystyrene insulation showed the highest change in cooling demand at 0.5%. However, the abovementioned change for the dry polystyrene was not highly significant, since it was less than 1%.

By contrast, the yearly average changes for cooling demand when applying different moisture levels of the polystyrene insulation, as observed from humid weather, were significant and need to be considered in future design processes. The cooling demand increased by 5% as the moisture level of polystyrene insulation increased by 10%. When the moisture level of the polystyrene content was doubled (20%), the change in cooling demand reached 6%. The highest change in required cooling demand was 6.5%, in the case of polystyrene with a 30% moisture content.

The results support the necessity of considering the change of the thermal conductivity of polystyrene as the moisture content changes due to humid hot weather in the designing process of cooling systems.

### 4.3. Yearly Cooling Capacity and Required Air Flow

The yearly cooling capacity and required air flow to control the room temperature within the comfortable temperature were assessed by employing variable thermal conductivity and compared with constant values of polystyrene at different levels of density and moisture content, as shown in Figure 6 and Figure 7, respectively.

Among the different polystyrene densities presented in Figure 6, LD polystyrene insulation resulted in the lowest yearly average change in cooling capacity and air supplied, at 0.55% and 0.73%, respectively. In the other hand, SHD polystyrene insulation showed the highest change in cooling capacity and supplied air at 0.73% and 1%, respectively.

The yearly average change in cooling capacity and required air flow at different moisture levels is presented in Figure 7. The yearly additional changes for cooling capacity and air flow at the 10% moisture level were 4.8% and 9.5%, respectively. Further increases of cooling capacity occurred as the moisture level increased to 20% of the polystyrene content, reaching 8.2%. The highest changes in cooling capacity and required air supplied were 8.9% and 11.2%, respectively, in the case of polystyrene with a 30% moisture content.

## 5. Conclusions

Accuracy of building energy assessment mainly depends on the accuracy of the overall heat transfer coefficient of the building envelope, which depends mainly on the thermal conductivity of the layers of the assembly, particularly the insulation material. In this study, the impact of changes in the thermal conductivity of EPS material was investigated by appling polystyrene insulation as part of a wall section with variable thermal conductivity (λ-value), subjected to yearly weather data of Al-Ain, UAE and compared to a constant thermal conductivity. The additional cooling demand and capacity due to the λ-relationship with time were assessed using e-quest as a building energy analysis tool. The results showed that when the λ-value was modeled as a function of operating temperature, its effect on the temperature profile during daytime was significant compared with cases of a constant λ-value. The yearly average change in space cooling demand and cooling capacity employing polystyrene with constant and variable thermal conductivity increased with the increase of the moisture content. Indeed, the highest changes in cooling demand and capacity were 6.5% and 8.8% with 30% moisture content polystyrene, highlighting the importance of taking the moisture level and operating temperature into account at the primary stage of building energy assessment for cooling system selection.

The current work had the limitation of evaluating the combined effect of temperature and moisture change for higher density levels. Therefore, in future studies measuring the dynamic hygrothermal response of the thermal conductivity of insulation and its impact on building energy performance, it would be worth employing a more appropriate model that takes into account the combined effect of temperature and moisture change. Moreover, similar studies should also be extended to other insulation materials, including fiberglass, mineral wool, cellulose, and polyurethane foam, which could be more sensitive to variations in the combined effect of temperature and humidity.

## Figures and Tables

**Figure 1 materials-12-03000-f001:**
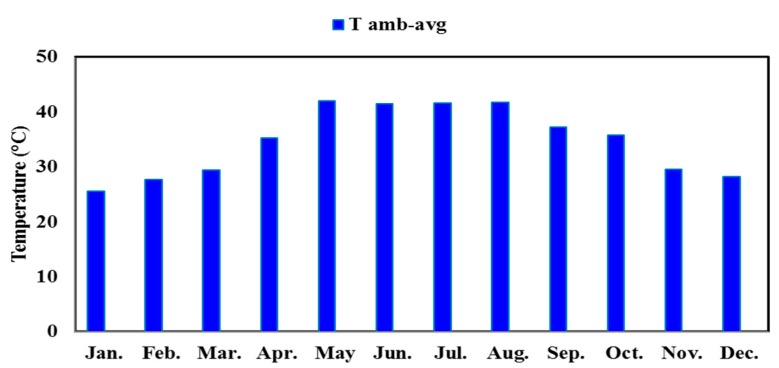
Average ambient temperature of Al-Ain, UAE for a typical day of each month.

**Figure 2 materials-12-03000-f002:**
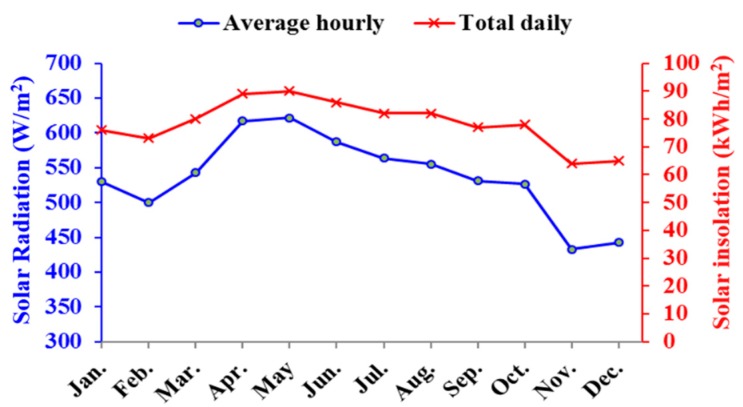
Average hourly and total solar radiation of Al-Ain, UAE for a typical day of each month.

**Figure 3 materials-12-03000-f003:**
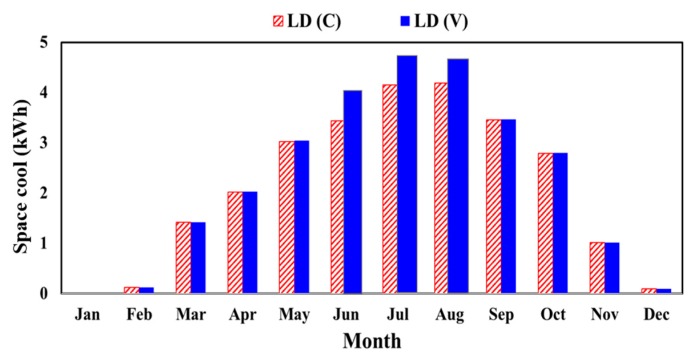
Required space cooling employing dry low-density polystyrene with constant (dashed bars) and variable (solid bars) thermal conductivity for residential buildings in hot climate of Al-Ain, UAE.

**Figure 4 materials-12-03000-f004:**
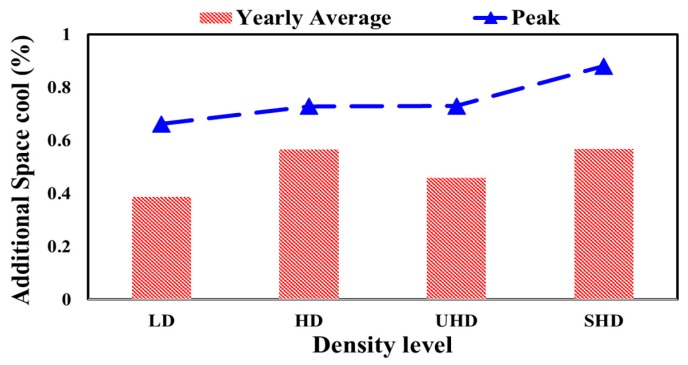
Yearly average and peak additional space cooling percentages employing dry variable thermal conductivity as compared with constant value at different density levels in the hot climate of UAE.

**Figure 5 materials-12-03000-f005:**
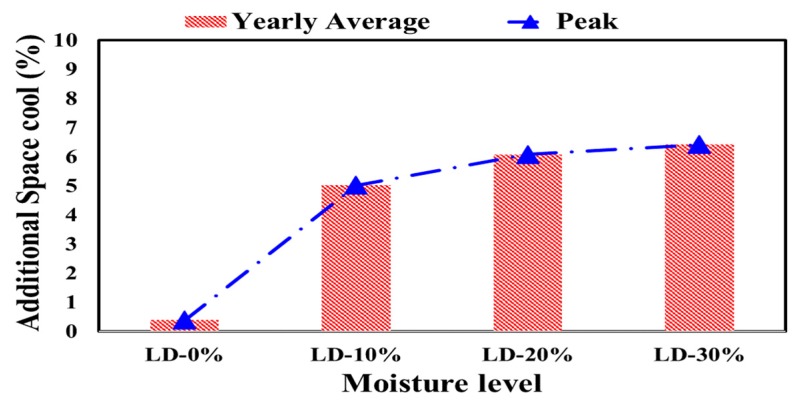
Yearly average and peak additional space cooling percentages employing dry variable thermal conductivity as compared with constant value at different moisture levels in the hot climate of UAE.

**Figure 6 materials-12-03000-f006:**
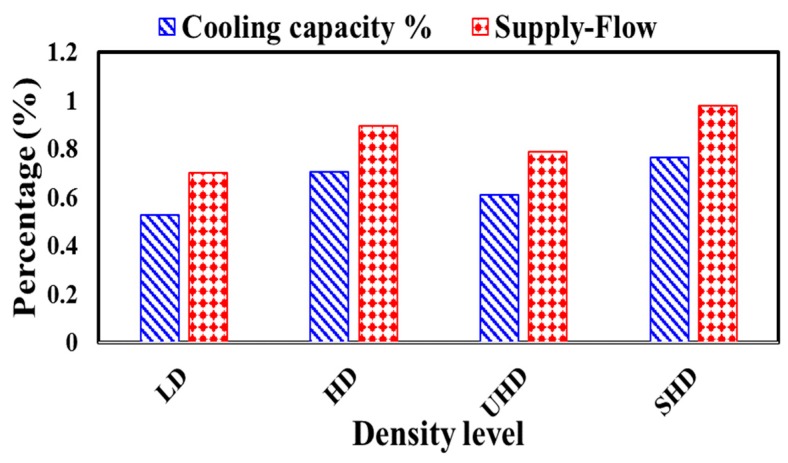
Yearly additional cooling capacity and supplied air flow (in percentages) by employing dry variable thermal conductivity as compared with constant values at different density levels in the hot climate of UAE.

**Figure 7 materials-12-03000-f007:**
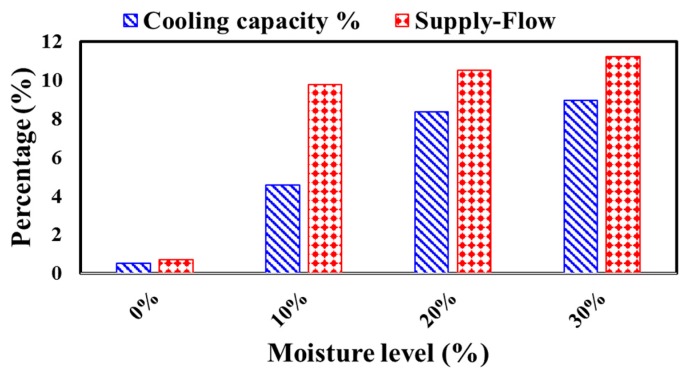
Yearly additional cooling capacity and supplied air flow (in percentages) by employing variable thermal conductivity as compared with constant values at different moisture levels in the hot climate of UAE.

**Table 1 materials-12-03000-t001:** Building characteristics and type of systems.

Characteristics	Description of the Base Case
Orientation	North
Height (Floor–Floor)	3.5 m
Floor Area	300 m^2^
Floor Dimension	20 × 15 m
Window Area	10% of the gross wall area, uniformly distributed
Window	6 mm single green-tinted glazing
	Thermal transmittance (U-value) = 5.788 W/m^2^·°C
	Solar heat gain coefficient (SHGC) = 0.623
Solar Absorbance	0.50 for external walls and roof
Wall	U-value = 2.388 W/m^2^·°C
Roof	U-value = 0.654 W/m^2^·°C
Floor	U-value = 0.781W/m^2^·°C
Occupancy Density	6 People
Lighting Power Density	4.5 W/m^2^
Equipment Power Density	7 W/m^2^

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
