# Peer review of "Yearly Energy Performance Assessment of Employing Expanded Polystyrene with Variable Temperature and Moisture–Thermal Conductivity Relationship"

_materials, 2019, doi:10.3390/ma12183000_

Round 1

Reviewer 1 Report

This is well presented paper which I should like to see published.

It is a calculations-only paper, and fairly straightforward in concept and execution.  I have no problem with a calculations-only paper provide that the question investigated is well-posed and important.

In the present case, I would ask the authors to justify the moisture contents of the insulation layer that they use. So far as I know, the PS insulation materials are entirely non-hygroscopic and have rather coarse pore structures. Therefore I would expect their moisture contents to be extremely small and insensitive to environmental humidity conditions.

A few points of detail:

The title is very wordy and obscure. Could the authors find a simple and clearer title?

Eqn 1 Could the authors please take care of the units?

In this the give values of Qin as for example kWh/day. so that in eqn 1 the time shouldn’t appear on the right-hand side.

Likewise in eqn 3, the value of 6.5 on the right-hand side implies certain units for hc, which should be stated.

The figures are well designed and the text is written in  good English.

The references are appropriate.

Subject to some minor revisions, I am happy to recommend this for publication.

Author Response

English language and style

( ) Extensive editing of English language and style required
( ) Moderate English changes required
(x) English language and style are fine/minor spell check required
( ) I don't feel qualified to judge about the English language and style

Yes

Can be improved

Must be improved

Not applicable

Does the introduction provide sufficient background and include all relevant references?

(x)

( )

( )

( )

Is the research design appropriate?

(x)

( )

( )

( )

Are the methods adequately described?

(x)

( )

( )

( )

Are the results clearly presented?

(x)

( )

( )

( )

Are the conclusions supported by the results?

(x)

( )

( )

( )

Comments and Suggestions for Authors

This is well presented paper which I should like to see published.

It is a calculations-only paper, and fairly straightforward in concept and execution.  I have no problem with a calculations-only paper provide that the question investigated is well-posed and important.

In the present case, I would ask the authors to justify the moisture contents of the insulation layer that they use. So far as I know, the PS insulation materials are entirely non-hygroscopic and have rather coarse pore structures. Therefore I would expect their moisture contents to be extremely small and insensitive to environmental humidity conditions.

Done. Please see page 3 paragraph 1.

A few points of detail:

The title is very wordy and obscure. Could the authors find a simple and clearer title?

The impact of the temperature and the moisture as combined effect on the energy performance of EPS insulation should appear in the title. We think that the title is apprpriate.

Eqn 1 Could the authors please take care of the units?

Done

In this the give values of Qin as for example kWh/day. so that in eqn 1 the time shouldn’t appear on the right-hand side.

Done

Likewise in eqn 3, the value of 6.5 on the right-hand side implies certain units for hc, which should be stated.

Done

The figures are well designed and the text is written in  good English.

The references are appropriate.

Subject to some minor revisions, I am happy to recommend this for publication.

Reviewer 2 Report

In my opinion, the paper cannot be accepted in the present form.

Technical and Quality

The paper has not a good scientific quality level and it does not present sufficient innovative aspects. The approach is not very rigorous and correct both for theoretical and formal aspects. In particular:

The model used for the heat demand calculation is very simple and it seems not to take into account any dynamic effects. I recommend to better describe the model and to include inertial effects; The building chosen in the study case is not representative of the stock building and it is not properly documented. Thus, the numerical results are not particularly significant. I recommend to justify the choice with respect to the typical stock building in your country; There isn’t any experimental validation of the obtained results. It is necessary to measure the effective energy consumption to validate your calculation.

The language is quite clear but it can be improved.

Presentation

About the presentation:

the title and abstract, they are adequate and they contain the essential information of the article; the reported literature about the temperature and particularly moisture influence on the conductivity is very poor and quite obsolete; The influence of moisture conductivity is not described in sufficient details considering the topic analysed by the author.

Author Response

The paper has not a good scientific quality level and it does not present sufficient innovative aspects. The approach is not very rigorous and correct both for theoretical and formal aspects. In particular:

The model used for the heat demand calculation is very simple and it seems not to take into account any dynamic effects. I recommend to better describe the model and to include inertial effects; The building chosen in the study case is not representative of the stock building and it is not properly documented. Thus, the numerical results are not particularly significant. I recommend to justify the choice with respect to the typical stock building in your country; There isn’t any experimental validation of the obtained results. It is necessary to measure the effective energy consumption to validate your calculation.

Answers

·         The details of the model which indeed takes the dynamic effect is included in the paper. Please see page 3 (Heat transfer Analysis).

·         The model of the house is well detailed in Table 1. This model has been already used in previous papers. Please see references 1, 10, 11, and 12.

·         The measurement of the thermal conductivity of different sample at various operating temperature and moisture contents has been carried out and already published in previous papers. Please see refs 1 and 10.

The language is quite clear but it can be improved.

Presentation

About the presentation:

the title and abstract, they are adequate and they contain the essential information of the article; the reported literature about the temperature and particularly moisture influence on the conductivity is very poor and quite obsolete; The influence of moisture conductivity is not described in sufficient details considering the topic analysed by the author.

·         Seven references have been added in the paper as requested by the reviewer.

·         A paragraph dealing with the moisture effect is added in page 3 as requested by the reviewer.

Reviewer 3 Report

The paper topic is interesting since discusses a case, but needs major revisions to be considered for a complete review. 

The introduction is based on many old dated references. This section should be updated by citing papers published in 2018 and 2019. This section should clarify aims and objectives, the significance of them, clear contributions.

Insert a literature review section before section 2. The paper cited 22 papers in the introduction section, and no more! Other sections also need specific references. For example, methods should be justified with methodology papers. 

In the discussion section, all results should be discussed citing similar or relevant articles published recently in terms of similarity or differences of your findings. 

The conclusion section also should be expanded and present the contributions, novelty, limitations and future directions.  At present, it looks like an industry report saying what is the highest change in the case selected. 

Author Response

The paper topic is interesting since discusses a case, but needs major revisions to be considered for a complete review. 

The introduction is based on many old dated references. This section should be updated by citing papers published in 2018 and 2019. This section should clarify aims and objectives, the significance of them, clear contributions.

Seven recent publications have been added in the paper.

Insert a literature review section before section 2. The paper cited 22 papers in the introduction section, and no more! Other sections also need specific references. For example, methods should be justified with methodology papers. 

Done. Please check 1.1 and 1.2

In the discussion section, all results should be discussed citing similar or relevant articles published recently in terms of similarity or differences of your findings. 

Done.

The conclusion section also should be expanded and present the contributions, novelty, limitations and future directions.  At present, it looks like an industry report saying what is the highest change in the case selected. 

Conclusion has been improved.

Round 2

Reviewer 2 Report

In my opinion, the paper cannot be accepted in the present form.

Although some improvements have been made, the paper does not yet have a good level of scientific quality and does not present sufficient innovative aspects.

Author Response

The references and the conclusion have been improved.